# KNOWLEDGE DISTILLATION VIA SOFTMAX REGRESSION REPRESENTATION LEARNING

**Jing Yang**
University of Nottingham
Nottingham, UK
`jing.yang2@nottingham.ac.uk`

**Brais Marinez**
Samsung AI Center
Cambridge, UK
`brais.mart@gmail.com`

**Adrian Bulat**
Samsung AI Center
Cambridge, UK
`adrian@adrianbulat.com,`

**Georgios Tzimiropoulos**
Samsung AI Center
Cambridge, UK
Queen Mary University of London
London, UK
`g.tzimiropoulos@qmul.ac.uk`

## ABSTRACT

This paper addresses the problem of model compression via knowledge distillation. We advocate for a method that optimizes the output feature of the penultimate layer of the student network and hence is directly related to representation learning. To this end, we *firstly* propose a direct feature matching approach which focuses on optimizing the student's penultimate layer only. *Secondly and more importantly*, because feature matching does not take into account the classification problem at hand, we propose a second approach that decouples representation learning and classification and utilizes the teacher's pre-trained classifier to train the student's penultimate layer feature. In particular, for the same input image, we wish the teacher's and student's feature to produce the same output when passed through the teacher's classifier, which is achieved with a simple $L_2$ loss. Our method is extremely simple to implement and straightforward to train and is shown to consistently outperform previous state-of-the-art methods over a large set of experimental settings including different (a) network architectures, (b) teacher-student capacities, (c) datasets, and (d) domains. The code is available at `https://github.com/jingyang2017/KD_SRRL`.

## 1 INTRODUCTION

Recently, there has been a great amount of research effort to make Convolutional Neural Networks (CNNs) lightweight so that they can be deployed in devices with limited resources. To this end, several approaches for model compression have been proposed, including network pruning (Han et al., 2016; Lebedev & Lempitsky, 2016), network quantization (Rastegari et al., 2016; Wu et al., 2016), knowledge transfer/distillation (Hinton et al., 2015; Zagoruyko & Komodakis, 2017), and neural architecture search (Zoph & Le, 2017; Liu et al., 2018). Knowledge distillation (Buciluǎ et al., 2006; Hinton et al., 2015) aims to transfer knowledge from one network (the so-called "teacher") to another (the so-called "student"). Typically, the teacher is a high-capacity model capable of achieving high accuracy, while the student is a compact model with much fewer parameters, thus also requiring much less computation. The goal of knowledge distillation is to use the teacher to improve the training of the student and push its accuracy closer to that of the teacher.

The rationale behind knowledge distillation can be explained from an optimization perspective: there is evidence that high capacity models (i.e. the teacher) can find good local minima due to over-parameterization (Du & Lee, 2018; Soltanolkotabi et al., 2018). In knowledge distillation, such models are used to facilitate the optimization of lower capacity models (i.e. the student) during training. For example, in the seminal work of (Hinton et al., 2015), the softmax outputs of the teacher provide extra supervisory signals of inter-class similarities which facilitate the training of

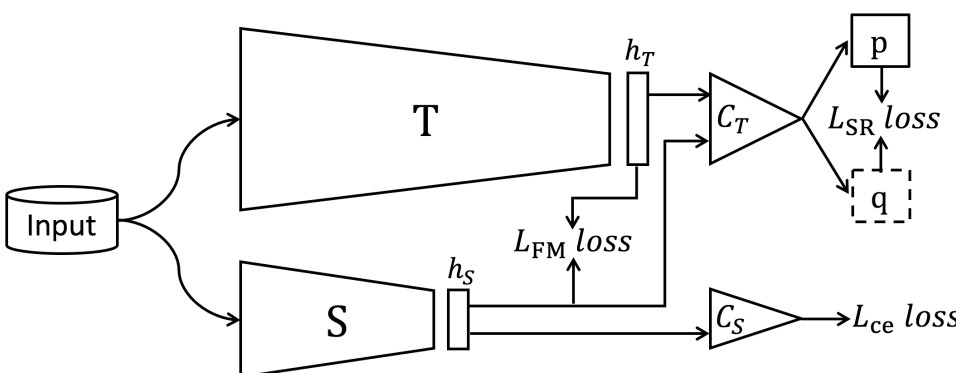

Figure 1: Our method performs knowledge distillation by minimizing the discrepancy between the penultimate feature representations $h_T$ and $h_S$ of the teacher and the student, respectively. To this end, we propose to use two losses: (a) the Feature Matching loss $L_{FM}$, and (b) the so-called Softmax Regression loss $L_{SR}$. In contrary to $L_{FM}$, our main contribution, $L_{SR}$, is designed to take into account the classification task at hand. To this end, $L_{SR}$ imposes that for the same input image, the teacher's and student's feature produce the same output when passed through the teacher's *pre-trained and frozen* classifier. Note that, for simplicity, the function for making the feature dimensionality of $h_T$ and $h_S$ the same is not shown.

the student. In other influential works, intermediate representations extracted from the teacher such as feature tensors (Romero et al., 2015) or attention maps (Zagoruyko & Komodakis, 2017) have been used to define auxiliary loss functions used in the optimization of the student.

Training a network whose output feature representation is rich and powerful has been shown crucial for achieving high accuracy for the subsequent classification task in recent works in both unsupervised and supervised learning, see for example (Chen et al., 2020; He et al., 2020) and (Kang et al., 2020). Hence, in this paper, we are advocating for representation learning-based knowledge distillation by optimizing the student's penultimate layer output feature. If we are able to do this effectively, we expect (and show experimentally) to end up with a student network which can generalize better than one trained with logit matching as in the KD paper of (Hinton et al., 2015).

**Main contributions:** To accomplish the aforementioned goal we propose two loss functions: *The first loss function*, akin to (Romero et al., 2015; Zagoruyko & Komodakis, 2017), is based on direct feature matching but focuses on optimizing the student's penultimate layer feature only. Because direct feature matching might be difficult due to the lower representation capacity of the student and, more importantly, is detached from the classification task at hand, we also propose *a second loss function*: we propose to decouple representation learning and classification and utilize the teacher's pre-trained classifier to train the student's penultimate layer feature. In particular, for the same input image, we wish the teacher's and student's feature to produce the same output when passed through the *teacher's* classifier, which is achieved with a simple $L_2$ loss (see Fig. 1). This softmax regression projection is used to retain from the student's feature the information that is relevant to classification, but since the projection matrix is pre-trained (learned during the teacher's training phase) this does not compromise the representational power of the student's feature.

**Main results:** Our method has two advantages: (1) It is simple and straightforward to implement. (2) It consistently outperforms state-of-the-art methods over a large set of experimental settings including different (a) network architectures (WideResNets, ResNets, MobileNets), (b) teacher-student capacities, (c) datasets (CIFAR-10/100, ImageNet), and (d) domains (real-to-binary).

## 2 RELATED WORK

**Knowledge transfer:** In the work of (Hinton et al., 2015), knowledge is defined as the teacher's outputs after the final softmax layer. The softmax outputs carry richer information than one-hot labels because they provide extra supervision signals in terms of the inter-class similarities learned

by the teacher. In a similar fashion to (Hinton et al., 2015), intermediate representations extracted from the teacher such as feature tensors (Romero et al., 2015) or attention maps (Zagoruyko & Komodakis, 2017) have been used to define loss functions used to facilitate the optimization of the student. Trying to match the whole feature tensor, as in FitNets (Romero et al., 2015), is hard and, in certain circumstances, such an approach may adversely affect the performance and convergence of the student. To relax the assumption of FitNet, Attention Transfer (AT) was proposed in (Zagoruyko & Komodakis, 2017) where knowledge takes the form of attention maps which are summaries of the energies of the feature tensors over the channel dimension. An extension of (Zagoruyko & Komodakis, 2017) using Maximum Mean Discrepancy of the network activations as a loss term for distillation was proposed in (Huang & Wang, 2017). Cho & Hariharan (2019) showed that very accurate networks are "too good" to be good teachers and proposed to mitigate this with early stopping of the teacher's training. Recently, the work of (Heo et al., 2019a) studied the location within the network at which feature distillation should be applied and proposed margin ReLU and a specifically designed distance function that transfers only the useful (positive) information from the teacher to the student. More recently, Li *et al.* (Li et al., 2020a) proposed to supervise the block-wise architecture search by the architecture knowledge distilled from a teacher model. Another NAS based method was proposed in (Guan et al., 2020), in which a student-to-teacher loss is used to find the aggregation weights that match the learning ability of the student. Passalis et al. (2020) claimed that traditional KD ignores information plasticity during the training process, and proposed to model the information flow through the various layers of the teacher.

**Feature relationship transfer:** Another line of knowledge distillation methods focus on exploring transferring the relationship between features, rather than the actual features themselves. In (Yim et al., 2017), feature correlations are captured by computing the Gram matrix of features across layers for both teacher and student and then applying an $L_2$ loss on pairs of teacher-student Gram matrices. The limitation of this work is the high computational cost, which is addressed to some extent in (Lee et al., 2018) by compressing the feature maps by singular value decomposition. Park et al. (2019) proposed a relational knowledge distillation method which computes distance-wise and angle-wise relations of each embedded feature vector. This idea is further explored in (Peng et al., 2019) and (Liu et al., 2019). In (Peng et al., 2019), Taylor series expansion is proposed to better capture the correlation between multiple instances. In (Liu et al., 2019), the instance feature and relationships are considered as vertexes and edges respectively in a graph and instance relationship graph is proposed to model the feature space transformation across layers. Inspired by the observation that semantically similar inputs should have similar activation patterns, (Tung & Mori, 2019) proposed a similarity-preserving knowledge distillation method which guides the student to mimic the teacher with respect to generating similar or dissimilar activations. More recently, (Jain et al., 2020) proposed to matching the student output with the teacher's by distilling the knowledge through a quantized visual words space. Li et al. (2020b) proposed the local correlation exploration framework to represent the relationships of local regions in the feature space which contains more details and discriminative patterns.

Finally, a similar connection between distillation and representation learning was very recently made in (Tian et al., 2020) which uses contrastive learning for knowledge distillation. We note that our loss is not related to the one used in (Tian et al., 2020), is simpler, and as shown in Section 5, outperforms it for all of our experiments, often by a significant margin.

## 3 METHOD

We denote by $T$ and $S$ the teacher and student networks respectively. We split these networks into two parts: (i) A convolutional feature extractor $f_{Net}, Net = \{T, S\}$, the output of which at the $i$-th layer is a feature tensor $F_{Net}^i \in \mathbb{R}^{C_{Net}^i \times H^i \times W^i}$, where $C_{Net}^i$ is the output feature dimensionality, and $H^i, W^i$ the output spatial dimensions. We also denote by $h_{Net} = \sum_{h=1}^{H^L} \sum_{w=1}^{W^L} F_{Net}^L \in \mathbb{R}^{C_{Net}^L}$ the last layer feature representation learned by $f_{Net}$. (ii) A projection matrix $W_{Net} \in \mathbb{R}^{C_{Net}^L \times K}$ which projects the feature representation $h_{Net}$ into $K$ class logits $z_{Net}^i, i = 1, \ldots, K$, followed by the `softmax` function $s(z_{Net}^i) = \frac{\exp(z_{Net}^i/\tau)}{\sum_j \exp(z_{Net}^j/\tau)}$ with temperature $\tau$ ($\tau = 1$ for Cross Entropy loss) which put together form a softmax regression classifier into $K$ classes.

Knowledge Distillation (KD) (Hinton et al., 2015) trains the student with the following loss:

$$L_{KD} = -\sum_{k=1}^{K} s(z_T^k) \log s(z_S^k), \qquad (1)$$

so that the discrepancy between the teacher's and student's classifiers is directly minimized.

FitNets (Romero et al., 2015) match intermediate feature representations. For the $i$-th layer, the following loss is defined:

$$L_{Fit} = \left\| F_T^i - r(F_S^i) \right\|^2, \qquad (2)$$

where $r(.)$ is a function for matching the feature tensor dimensions.

In our work, we propose to minimize the discrepancy between the representations $h_T$ and $h_S$. To accomplish this goal, we propose to use two losses. The first one is an $L_2$ feature matching loss:

$$L_{FM} = \|h_T - h_S\|^2, \qquad (3)$$

where for notational simplicity we dropped the dependency on $r(.)$. Hence, $L_{FM}$ loss is a simplified FitNet loss which focuses only on the final representation learned. The intuition for this is that this feature is directly connected to the classifier and hence imposing the student's feature to be similar to that of the teacher could have more impact on classification accuracy. Moreover, it might be questionable why one should optimize for other intermediate representations as in (Romero et al., 2015) especially when the student is a network of lower representational capacity. In Section 4: *Where should the losses be applied?*, we confirm that $L_{FM}$ alone has a positive impact but feature matching in other layers is not helpful.

We found $L_{FM}$ to be effective but only to limited extent. One disadvantage of $L_{FM}$ and, in general, of all feature matching losses e.g. (Romero et al., 2015; Zagoruyko & Komodakis, 2017), is that it treats each channel dimension in the feature space independently, and ignores the inter-channel dependencies of the feature representations $h_S$ and $h_T$ for the final classification. This is in contrast to the original logit matching loss proposed by Hinton *et al.* in (Hinton et al., 2015) which directly targets classification accuracy. To alleviate the aforementioned problem, in this work, we propose a second loss for optimizing $h_S$ which is directly linked with classification accuracy. To this end, we will use the teacher's *pre-trained* Softmax Regression (SR) classifier.

Let us denote by $p$ the output of the teacher network when fed with some input image $x$. Let us also feed the same image through the student network to obtain feature $h_S(x)$. Finally let us pass $h_S(x)$ through the teacher's SR classifier to obtain output $q$. See also Fig. 1. Our loss is defined as:

$$L_{SR} = -p \log q. \qquad (4)$$

At this point, we make the following two observations: (1) If $p = q$ (and since the teacher's classifier is frozen), then this implies that $h_S(x) = h_T(x)$ which shows that indeed Eq. (4) optimizes the student's feature representation $h_S$ ($h_T$ is also frozen). (2) The loss of Eq.(4) can be written as:

$$L_{SR} = -s(W_T' h_T) \log s(W_T' h_S). \qquad (5)$$

Now let us now write KD loss in a similar way:

$$L_{KD} = -s(W_T' h_T) \log s(W_s' h_S). \qquad (6)$$

By comparing Eq. (5) with Eq. (6), we see that the only difference in our method is that the frozen, pre-trained teacher's classifier is used for both teacher and the student. On the contrary, in KD, $W_S$ is also optimized. This gives more degrees of freedom to the optimization algorithm, in particular, to adjust the weights of *both* the student's feature extractor $f_S$ *and* the student's classifier $W_S$ in order to minimize the loss. This has an impact on the learning of the student's feature representation $h_S$ which, in turn, hinders the generalization capability of the student on the test set. We confirm this hypothesis with the experiment of Section 4: *Transferability of representations*.

Finally, we note that we found that, in practice, an $L_2$ loss between the logits:

$$L_{KD} = \left\| W_T' h_T - W_T' h_S \right\|^2 = \|h_T - h_S\|_{W_T}^2, \qquad (7)$$

works slightly better than the cross-entropy loss. The comparison between different types of losses for $L_{SR}$ is given in the appendix.

Table 1: Effect of proposed losses ($L_{FM}$ and $L_{SR}$) and position of distillation on the test set of CIFAR-100.

| Method | Layer | Top-1 (%) | Top-5 (%) |
|---|---|---|---|
| Student (WRN-16-4) | | 76.97 | 93.89 |
| Teacher (WRN-40-4) | | 79.50 | 94.57 |
| $L_{FM}$ | conv4 | 78.05 | 94.45 |
| $L_{SR}$ | conv4 | 79.10 | 94.99 |
| $L_{FM}+L_{SR}$ | conv4 | **79.58** | **95.21** |
| $L_{FM}+L_{SR}$ | conv2 | 77.03 | 93.94 |
| $L_{FM}+L_{SR}$ | conv3 | 77.34 | 94.22 |
| $L_{FM}+L_{SR}$ | conv2+3+4 | 79.43 | 94.80 |

Overall, in our method, we train the student network using three losses:

$$L = L_{CE} + \alpha L_{FM} + \beta L_{SR}, \tag{8}$$

where $\alpha$ and $\beta$ are the weights used to scale the losses. The teacher network is pretrained and fixed during training the student. $L_{CE}$ is the standard loss based on ground truth labels for the task in hand (e.g. cross-entropy loss for image classification). Note that this results in a very simple algorithm for training the student, summarized in Algorithm 1.

---

**Algorithm 1** Knowledge distillation via Softmax Regression Representation Learning

---

**Input:** Teacher network $T$, Student network $S$, input image $\mathbf{x}$, ground truth label $y$, weights $\alpha$, $\beta$.
1. Input $\mathbf{x}$ to $S$ to obtain feature $h_S$ and class prediction $\hat{y}$. Calculate cross entropy loss $L_{CE} = \mathcal{H}(\hat{y}, y)$;
2. Input $\mathbf{x}$ to $T$ to obtain feature $h_T$. Calculate distillation losses from Eqs. (3) and (7);
3. Update $S$ by optimizing Eq. (8)

**Output:** the updated $S$.

---

## 4 ABLATION STUDIES

We conducted a set of ablation studies on CIFAR-100 (see Section 5.1) using a Wide ResNet (WRN) for both teacher (WRN-40-4) and student (WRN-16-4) (for network definitions, see Section 5).

**Are both $L_{FM}$ and $L_{SR}$ useful?** To answer this question, we ran 3 experiments: using $L_{FM}$ alone, $L_{SR}$ alone, and combining them together $L_{FM} + L_{SR}$. The results of Table 1 (first 3 rows) clearly show that all proposed variants offer significant gains: when using $L_{FM}$ and $L_{SR}$ alone, $\sim 1\%$ and $\sim 2\%$ improvements in Top-1 accuracy were obtained. Moreover, when combining them together, an additional $\sim 0.4\%$ improvement was gained. Importantly, the results show that $L_{SR}$ is significantly more effective than $L_{FM}$. We further note at this point that we found that $L_{FM}$ offers diminishing gains on ImageNet experiments.

**Where should the losses be applied?** The proposed losses can be applied at other layers of the networks too. This is straightforward for $L_{FM}$. We can also extend $L_{SR}$ to more layers, by transferring the mean feature of the student at each layer to the corresponding layer of the teacher using an AdaIN layer (Huang & Belongie, 2017). On one hand, applying the losses early in the network could ensure that the subsequent layers receive "better" features. On the other hand, features produced by early layers are not specialised to a particular class. Thus, applying the distillation losses towards the end of the network, where the activations encode discriminative, task-related features should lead to potentially stronger models. The results from Table 1 (last 3 rows) confirm our hypothesis: Applying the loss at multiple points in the network actually rather hurts accuracy.

**Teacher-student similarity:** The overall aim of knowledge distillation is to make the student mimic the teacher's output, so that the student is able to obtain similar performance to that of the teacher. Therefore, to see how well the student mimics the teacher, we measured the similarity between the teacher's and student's outputs using (a) the KL divergence between the teacher's and student's outputs, and (b) the cross-entropy loss between the student's predictions and the ground truth labels.

Table 2: KL divergence between teacher and student, and cross-entropy between student and ground truth on the test set of CIFAR-100. Teacher's top-1 accuracy is 79.50%.

| Method | KL div. with teacher | Cross-entropy with label | Top-1 (%) |
|---|---|---|---|
| Student | 0.5964 | 0.9383 | 76.97 |
| KD | 0.5818 | 0.9492 | 78.35 |
| AT | 0.5406 | 0.9049 | 78.06 |
| $L_{FM}$ | 0.5701 | 0.8980 | 78.05 |
| $L_{SR}$ | 0.4828 | 0.8418 | 79.10 |
| $L_{FM}+L_{SR}$ | **0.4597** | **0.8247** | **79.58** |

Table 3: $L_2$ Distance $\|h_T - h_S\|^2$ , and NMI calculated on the test set of CIFAR-100.

| Method | Student | $L_{FM}$ | $L_{SR}$ | $L_{FM}+L_{SR}$ |
|---|---|---|---|---|
| L2 Distance | 1.48 | 1.33 | 1.07 | **1.01** |
| NMI (%). | 77.20 | 78.31 | 79.35 | **79.85** |
| Top-1(%). | 76.97 | 78.05 | 79.10 | **79.58** |

From Table 2, it can be observed that KD (Hinton et al., 2015) reduces the KL divergence with the teacher's output offering $\sim 1.5\%$ accuracy gain. AT (Zagoruyko & Komodakis, 2017) also decreases the KL divergence with the teacher's output offering a smaller accuracy gain of $\sim 1.0\%$. Moreover, both proposed losses $L_{FM}$ and $L_{SR}$ and their combination $L_{FM}+L_{SR}$ show considerably high similarity compared to the KD and AT. This similarity is one of the main reasons for the improved student's accuracy offered by our method.

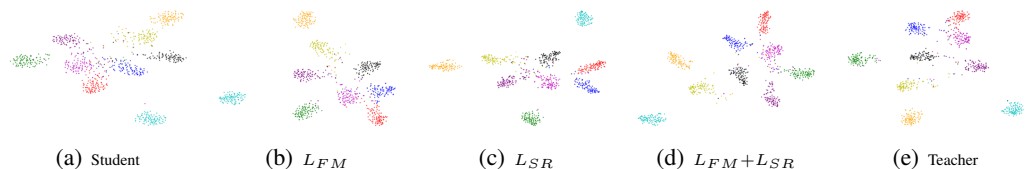

(a) Student     (b) $L_{FM}$     (c) $L_{SR}$     (d) $L_{FM}+L_{SR}$     (e) Teacher

Figure 2: Visualization of $h_S$ and $h_T$ on the test set of CIFAR-100. Better viewed in color.

**Representations distance:** Table 3 shows the $L_2$ distance between the teacher and student representations $h_T$ and $h_S$. The results, presented in Table 3, clearly show that both $L_{FM}$ and $L_{SR}$ narrow the distance, with their combination being the closest to the teacher.

**Normalized Mutual Information (NMI):** Moreover, we calculated the NMI (Manning et al., 2008) which is a balanced metric that can be used to determine the quality of feature clustering. The results, presented in Table 3, show that $L_{FM} + L_{SR}$ has the highest NMI score, meaning that the features are better clustered. Qualitative results are shown in Figure 2, which visualizes the features $h_S$ and $h_T$. It can be observed that $L_{FM}+L_{SR}$ is able to learn more discriminative features, which also correlates with quantitative accuracy gains.

**Transferability of representations:** Following (Tian et al., 2020), this section aims to compare the representational power of the learned student's representation $h_S$. To this end, we trained the student on CIFAR100, and then used it as a frozen feature extractor on top of which we train a linear classifier for 2 datasets: STL10 Coates et al. (2011) and CIFAR100. We compare the transfer ability of KD, CRD, $L_{FM}$, $L_{SR}$, and $L_{FM}+L_{SR}$. The superiority of the proposed losses over KD on STL is evident. Importantly, $L_{SR}$ largely outperforms KD which confirms our analysis of Eqs. (5) and (6). The best results on STL are obtained by CRD. However, on CIFAR100, which is the target distillation dataset our method outperforms CRD.

## 5   COMPARISON WITH STATE-OF-THE-ART

We thoroughly evaluated the effectiveness of our method across multiple (a) network architectures (ResNet (He et al., 2016), Wide ResNet (Zagoruyko & Komodakis, 2016), MobileNetV2 (Sandler et al., 2018), MobileNet (Howard et al., 2017)) with different teacher-student capacities, (b) datasets

Table 4: Transferability of representations from CIFAR100 to STL-10 and CIFAR100 by freezing $f^S$ and training a linear classifier on top. Top 1 (%) accuracy is provided.

| Student | Dataset | KD | CRD | $L_{FM}$ | $L_{SR}$ | $L_{FM}+L_{SR}$ |
|---------|---------|-------|---------|----------|----------|-----------------|
| WRN16-4 | STL10 | 68.75 | **72.45** | 69.3 | 71.44 | 72.17 |
| WRN16-4 | CIFAR100 | 78.28 | 78.46 | 77.95 | 79.03 | **79.34** |
| MobileNetV2 | STL10 | 62.17 | **69.74** | 66.12 | 68.23 | 68.95 |
| MobileNetV2 | CIFAR100 | 69.17 | 70.68 | 70.66 | 71.00 | **71.63** |

Table 5: Top-1 accuracy (%) of various knowledge distillation methods on CIFAR-10.

| Student(Params) | Teacher(Params) | Student | KD | AT | OFD | RKD | Ours | Teacher |
|-----------------|-----------------|---------|-------|-------|-------|-------|---------|---------|
| WRN-16-1 (0.18M) | WRN-16-2 (0.69M) | 91.04 | 92.57 | 92.15 | 92.28 | 92.51 | **92.95** | 93.98 |
| WRN-16-2 (0.69M) | WRN-40-2 (2.2M) | 93.98 | 94.46 | 94.39 | 94.30 | 94.41 | **94.66** | 95.07 |
| ResNet-8 (0.08M) | ResNet-26 (0.37M) | 87.78 | 88.75 | 88.15 | 87.49 | 88.50 | **89.02** | 93.58 |
| ResNet-14 (0.17M) | ResNet-26 (0.37M) | 91.59 | 92.57 | 92.11 | 92.51 | 92.36 | **92.70** | 93.58 |
| ResNet-18 (0.7M) | ResNet-34 (1.4M) | 93.35 | 93.74 | 93.52 | 93.80 | 92.95 | **93.92** | 94.11 |
| WRN-16-1 (0.18M) | ResNet-26 (0.37M) | 91.04 | 92.42 | 91.32 | 92.47 | 92.08 | **92.94** | 93.58 |

(CIFAR10/100, ImageNet), and (c) domains (real-valued and binary networks). The training details for all experiments are provided in the appendix. We denote with ResNet-N a Residual Network with N convolutional layers (He et al., 2016). We denote with WRN-$D$-$k$ a WRN architecture with $D$ layers and an expansion rate of $k$ (Zagoruyko & Komodakis, 2017).

For the above mentioned settings, we compare our method with KD (Hinton et al., 2015) and AT (Zagoruyko & Komodakis, 2017), and the more recent methods of OFD (Heo et al., 2019a), RKD (Park et al., 2019), CRD (Tian et al., 2020).

**Overview of results:** From our experiments, we conclude that our approach offers consistent gains across all of the above scenarios, outperforming all methods considered for all settings. Notably, our method is particularly effective for the most difficult datasets (i.e. CIFAR-100 and ImageNet).

### 5.1 CIFAR-10/100

For CIFAR-10, Top-1 performance of our method is shown in Table 5. We tested three cases representing different network architectures for student and teacher networks: the first two experiments are with WRNs. The following three experiments are with ResNets. In the last experiment, teacher and student have different network architectures. Overall, our method achieves the best results for all cases, with KD (Hinton et al., 2015) closely following.

For CIFAR-100 (Krizhevsky & Hinton, 2009), we experimented with several student-teacher network pairs using different structures. Experiments are grouped in three sets. The first shows performance for different teacher and student capacities using WRNs: poor student - good teacher (WRN-16-2; WRN-40-4), descent student - good teacher (WRN-10-10; WRN-16-10); good student - good teacher (WRN-16-4; WRN-40-4). In the second set, we show that these results hold when using a different architecture, ResNet in this case. The final set is designed to show performance when teacher and student have different architectures (MobileNetV2, ResNet and WRN).

Top-1 performance of our method is shown in Table 11. We observe that for almost all configurations, our method achieves consistent and significant accuracy gains over prior work. Furthermore, it is hard to tell which is the second best method as the remaining methods have their own advantages for different configurations. For WRN experiments, OFD ranks second. For ResNet and mixed structure experiments, CRD ranks second. More comparisons with other methods and results obtained by combining our method with KD and AT are provided in the supplementary material. Further improvements could be obtained by combining our method with others but this requires a comprehensive investigation which goes beyond the scope of this paper.

Table 6: Top-1 accuracy (%) of various knowledge distillation methods on CIFAR-100.

| Student (Params) | Teacher (Params) | Student | KD | AT | OFD | RKD | CRD | Ours | Teacher |
|---|---|---|---|---|---|---|---|---|---|
| WRN-16-2 (0.70M) | WRN-40-4 (8.97M) | 72.70 | 74.52 | 74.33 | 75.57 | 74.23 | 75.27 | **75.96** | 79.50 |
| WRN-16-4 (2.77M) | WRN-40-4 (8.97M) | 76.97 | 78.35 | 78.06 | 79.29 | 78.38 | 78.83 | **79.58** | 79.50 |
| WRN-10-10 (7.49M) | WRN-16-10 (17.2M) | 76.27 | 78.20 | 76.44 | 78.72 | 77.84 | 78.35 | **79.17** | 79.77 |
| ResNet-10 (0.34M) | ResNet-34 (1.39M) | 68.42 | 69.18 | 68.49 | 68.94 | 68.70 | **70.24** | 69.91 | 72.05 |
| ResNet-18 (0.75M) | ResNet-50 (1.99M) | 71.07 | 73.41 | 71.90 | 72.79 | 70.93 | 73.23 | **73.47** | 73.31 |
| ResNet-10 (4.95M) | ResNet-34 (21.33M) | 75.01 | 77.35 | 76.87 | 77.35 | 77.46 | 77.37 | **77.90** | 78.44 |
| WRN-16-2 (0.70M) | ResNet-34 (21.33M) | 72.70 | 73.95 | 72.32 | 74.78 | 73.91 | 74.88 | **75.38** | 78.44 |
| MobileNetV2 (2.37M) | ResNet-34 (21.33M) | 68.42 | 69.36 | 68.60 | 69.45 | 68.75 | 71.36 | **71.58** | 78.44 |
| MobileNetV2 (2.37M) | WRN-40-4 (8.97M) | 68.42 | 69.15 | 68.95 | 70.08 | 68.19 | 71.46 | **71.82** | 79.50 |

Table 7: Comparison with state-of-the-art on ImageNet.

| Student (Params) | Teacher (Params) | | Student | KD | AT | OFD | RKD | CRD | Ours | Teacher |
|---|---|---|---|---|---|---|---|---|---|---|
| ResNet18 (11.69M) | ResNet34 (21.80M) | Top-1 | 70.04 | 70.68 | 70.59 | 71.08 | 71.34 | 71.17 | **71.73** | 73.31 |
| | | Top-5 | 89.48 | 90.16 | 89.73 | 90.07 | 90.37 | 90.13 | **90.60** | 91.42 |
| MobileNet (4.23M) ) | ResNet50 (25.56M) | Top-1 | 70.13 | 70.68 | 70.72 | 71.25 | 71.32 | 71.40 | **72.49** | 76.16 |
| | | Top-5 | 89.49 | 90.30 | 90.03 | 90.34 | 90.62 | 90.42 | **90.92** | 92.86 |

## 5.2 IMAGENET-1K

Our experiments include two pairs of networks which are popular settings for ImageNet (Russakovsky et al., 2015). The first is distillation from ResNet-34 to ResNet-18 and the second one is distillation from ResNet-50 to MobileNet (Howard et al., 2017). Note that, following (Tian et al., 2020) on ImageNet, for KD, we set the weight for the KL loss to 0.9, the weight for cross-entropy loss to 0.5 which helps to obtain better accuracy.

Our results are presented in Table 7. Again, we observe that our method achieves significant improvements over all competing methods. Moreover, there is no method which is consistently second: for ResNet-34 to ResNet-18 experiment, RKD is the second best while for ResNet-50 to MobileNet, CRD is the second best. Notably, for the latter experiment, CRD reduces the gap between the teacher and the student by 1.27%, while our method narrows it by 2.36%. Overall, our results on ImageNet validate the scalability of our method, and show that, when applied to a large-scale dataset, we achieve an even more favourable performance compared against competing methods.

Table 8: Real-to-binary distillation results on CIFAR-100: a real-valued teacher ResNet-34 is used to distill a binary student. Real-to-binary distillation results on ImageNet-1K: a real-valued ResNet-18 is used to distill a binary student. OFD result might be suboptimal.

| Dataset | Method | Binary | KD | AT | OFD | RKD | CRD | Ours | Real |
|---|---|---|---|---|---|---|---|---|---|
| CIFAR-100 | ResNet34 | 65.34 | 68.65 | 68.54 | 66.84 | 68.61 | 68.78 | **70.50** | 75.08 |
| ImageNet-1K | ResNet18 | 56.70 | 57.39 | 58.45 | 55.74 | 58.84 | 58.25 | **59.57** | 70.20 |

## 5.3 BINARY NETWORKS DISTILLATION

Training highly accurate binary neural networks (i.e. the most extreme case of quantization) is a very challenging task (Rastegari et al., 2016; Bulat & Tzimiropoulos, 2019), and to this end, knowledge distillation appears to be a promising direction. In this section, we present results by applying distillation for the task of training binary student networks guided by real-valued teacher networks. The network architecture is kept the same for both the student and the teacher in this case: specifically we used a ResNet using the modifications described in (Bulat & Tzimiropoulos, 2019).

Table 8 presents our results. Again, we observe that our method outperforms all methods considerably, showing that it can effectively transfer knowledge between different domains. Note that it was not clear to us where to place the distillation position for OFD, so although we included our result for this method, we emphasize that this result might be suboptimal.

## 5.4 FACIAL LANDMARK DETECTION

Given a face image, the task is to localise a set of facial landmarks in terms of their (x,y) coordinates. This is often solved by using a CNN to directly regress the (x,y) coordinates of the facial landmarks. In order to show the suitability of our method for this problem, we use the WFLW Wu et al. (2018) dataset, which is one of the hardest benchmarks for this task. Performance is measured in terms of Normalised Mean Error (lower is better), which is the standard metric for the problem. In our experiment, we use a ResNet50 as the teacher and a ResNet8 as the student. The results, shown in Table 9, confirm the superior performance of our method when compared to other state-of-the-art methods.

Table 9: Facial landmark detection with ResNet50 as teacher and ResNet8 as student. KD is adapted by using an L2 loss instead of a KL loss to measure the discrepancy between the teach and student predictions.

| Student(Params) | Teacher(Params) | - | Student | KD | RKD | PKT | $L_{FM}$ | AT | Ours | Teacher |
|---|---|---|---|---|---|---|---|---|---|---|
| ResNet8(7.25M) | ResNet50(26.25M) | NME | 7.43 | 7.32 | 6.94 | 7.09 | 7.14 | 6.96 | **6.81** | 6.38 |

## 6 CONCLUSION

We presented a method for knowledge distillation that optimizes the output feature of the penultimate layer of the student network and hence is directly related to representation learning. A key to our method is the newly proposed Softmax Regression Loss which was found necessary for effective representation learning. We showed that our method consistently outperforms other state-of-the-art distillation methods for a wide range of experimental settings including multiple network architectures (ResNet, Wide ResNet, MobileNet) with different teacher-student capacities, datasets (CIFAR10/100, ImageNet), and domains (real-valued and binary networks).

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

## A  APPENDIX

### A.1  STUDY OF THE HYPER-PARAMETERS $\alpha$ AND $\beta$

We only performed a very basic search to find the best hyper-parameters. First, we fix $\alpha$ and search for the best $\beta$. Then, we used the best $\beta$, and search for the best $\alpha$. This is sub-optimal compared to a full grid search over $\alpha$ and $\beta$. Furthermore, after some preliminary experimentation, we considered only for 2 values: 1 and 5. Notably, we found that for all teacher-student pairs but (T:WRN40_4, S:MV2) alpha=1 is the optimal value. Furthermore, on ImageNet, the optimal values were $\alpha = 1$, $\beta = 1$ for all teacher-student pairs considered.

### A.2  DATASETS AND TRAINING DETAILS

**CIFAR-10**  CIFAR-10 is a popular image classification dataset consisting of 50,000 training and 10,000 testing images equally distributed across 10 classes. All images are of resolution $32 \times 32$px. Following (Zagoruyko & Komodakis, 2017), during training, we randomly cropped and horizontally flipped the images. The ResNet models were trained for 350 epochs using SGD. The initial learning rate was set to 0.1, and then it was reduced by a factor of 10 at epochs 150, 250 and 320. Similarly, the WRN models were trained for 200 epochs with a learning rate of 0.1 that was subsequently reduced by 5 at epochs 60, 120 and 160. In all experiments, we set the dropout rate to 0.

For traditional KD (Hinton et al., 2015), we set $\alpha = 0.9$ and $T = 4$. For AT (Zagoruyko & Komodakis, 2017), as in (Zagoruyko & Komodakis, 2017; Tung & Mori, 2019), we set the weight of distillation loss to 1000. We note that, in our experiments, the AT loss is added after each layer group for WRN and the last two groups for ResNet as in (Zagoruyko & Komodakis, 2017). Following OFD (Heo et al., 2019a), we set the weight of distillation loss to $10^{-3}$. For RKD (Park et al., 2019), we set $\beta_1 = 25$ for distance, and $\beta_2 = 50$ for angle, as described in (Park et al., 2019; Tian et al., 2020). We did not compare with CRD (Tian et al., 2020) on CIFAR-10 because, in our experiments, we found that their parameter setting (used in their paper for CIFAR-100 and ImageNet-1K) does not obtain good performance on CIFAR-10.

**CIFAR-100** For CIFAR-100 (Krizhevsky & Hinton, 2009), we used a standard data augmentation scheme (Zagoruyko & Komodakis, 2017) including padding 4 pixels prior to random cropping and horizontal flipping. We used SGD with weight decay 5e-4 and momentum 0.9. Batch size was set to 128. Learning rate was set to 0.1; then decayed by 0.1 at epochs 100, 150, until training reached 200 epochs (Heo et al., 2019a).

**ImageNet-1K** Images are cropped to $224 \times 224$ pixels for both training and evaluation. We used SGD with Nesterov momentum 0.9, weight decay $1e - 4$, initial learning rate 0.2 which was then dropped by a factor of 10 every 30 epochs, training in total for 100 epochs (for CRD we trained with 10 more epochs as suggested by the authors). Batch size was set to 512. For simplicity and to enable a fair comparison, we used pretrained PyTorch models Paszke et al. (2017) as teacher networks Heo et al. (2019a); Tian et al. (2020). For binary experiments, we used Adam as the optimizer with initial learning 0.002 which was then reduced by a factor of 10 every 30 epochs, training in total for 100 epochs.

## A.3 ADDITIONAL ABLATION STUDIES

**Different losses for $L_{SR}$:** This part expands Section 4 of our paper by evaluating different losses for $L_{SR}$. The following loss functions are compared:

1. L2 loss: $L_{SR-L2}(p, q) = \|p - q\|^2$. This is the loss used in Section 4 of our paper
2. Cross Entropy loss (CE) with label $y$: $L_{SR-CE}(q, y) = \mathcal{H}(q, y)$.
3. KL loss with temperature $\tau$ Hinton et al. (2015): $L_{SR-KL}(p, q) = KL(q/\tau, p/\tau)$.

The results, presented in Table 10, show that all loss functions offer significant improvement gains while $L_{FM}+L_{SR-L2}$ achieves the best accuracy. Therefore, in our paper, $L_{SR-L2}$ is used in all cases.

Table 10: Evaluation of different loss functions for $L_{SR}$ in terms of Top-1 accuracy on CIFAR-100.

| Method | Top-1(%) | Top-5(%) |
|---|---|---|
| Student: WRN-16-4 | 76.97 | 93.89 |
| Teacher: WRN-40-4 | 79.50 | 94.57 |
| $L_{FM}+L_{SR-L2}$ | **79.58** | **95.21** |
| $L_{FM}+L_{SR-CE}$ | 78.80 | 95.13 |
| $L_{FM}+L_{SR-KL}$ | 79.04 | 95.12 |

**Combining our method with KD and AT:** Table 11 shows additional comparisons on CIFAR100 by combining our method with AT Zagoruyko & Komodakis (2017) and KD Hinton et al. (2015), respectively. The results show that a straightforward combination did not provide satisfactory results, however it could be possible that a more comprehensive investigation might prove to be beneficial.

## A.4 ADDITIONAL COMPARISONS

This section provides additional comparisons using the evaluation framework of CRD Tian et al. (2020). Comparisons include distillation between models with the same architecture (e.g. ResNet56 to ResNet20) and between different architectures (e.g. ResNet50 to MobileNetV2). In order to

Table 11: Top-1 accuracy (%) of combining our method with KD and AT on CIFAR-100.

| Student (Params) | Teacher (Params) | Student | KD | AT | KD+Ours | AT+Ours | Ours | Teacher |
|---|---|---|---|---|---|---|---|---|
| WRN-16-2 (0.70M) | WRN-40-4 (8.97M) | 72.70 | 74.52 | 74.33 | 74.97 | 75.01 | **75.96** | 79.50 |
| WRN-16-4 (2.77M) | WRN-40-4 (8.97M) | 76.97 | 78.35 | 78.06 | 79.00 | 79.09 | **79.58** | 79.50 |
| WRN-10-10 (7.49M) | WRN-16-10 (17.2M) | 76.27 | 78.20 | 76.44 | 78.84 | 77.79 | **79.17** | 79.77 |
| ResNet-10 (0.34M) | ResNet-34 (1.39M) | 68.42 | 69.18 | 68.49 | **70.41** | 69.41 | 69.91 | 72.05 |
| ResNet-18 (0.75M) | ResNet-50 (1.99M) | 71.07 | 73.41 | 71.90 | 73.46 | 73.17 | **73.47** | 72.83 |
| ResNet-10 (4.95M) | ResNet-34 (21.33M) | 75.01 | 77.35 | 76.87 | 77.64 | 77.48 | **77.90** | 78.44 |
| WRN-16-2 (0.70M) | ResNet-34 (21.33M) | 72.70 | 73.95 | 72.32 | 74.90 | 74.71 | **75.38** | 78.44 |
| MobileNetV2 (2.37M) | ResNet-34(21.33M) | 68.42 | 69.36 | 68.60 | 71.08 | 70.70 | **71.58** | 78.44 |
| MobileNetV2 (2.37M) | WRN-40-4 (8.97M) | 68.42 | 69.15 | 68.95 | 70.85 | 70.63 | **71.82** | 79.50 |

maximize the fairness of the comparison, we followed their experimental setting. Thus, we did not choose the training parameters, teacher-student architecture pairs or methods to compare against. The competing methods included are:

- *Classic:* Knowledge Distillation (KD) Hinton et al. (2015), FitNet Romero et al. (2015), Attention Transfer (AT) Zagoruyko & Komodakis (2017).

- *Most recent:* Similarity-Preserving KD (SP) (Tung & Mori, 2019), Correlation Congruence (CC) (Peng et al., 2019), Variational Information Distillation (VID) (Ahn et al., 2019), Relational Knowledge Distillation (RKD) (Park et al., 2019), Distillation of Activation Boundaries (AB) (Heo et al., 2019b), Factor Transfer (FT) (Kim et al., 2018), Flow of Solution (FSP) (Yim et al., 2017) and Contrastive Representation Distillation (CRD) (Tian et al., 2020).

Table 12: **Distillation experiment with the same architectures (Tian et al., 2020):** Top-1 accuracy (%) on CIFAR-100. The student models were trained with a teacher of the same architecture. We report average over 3 runs as in (Tian et al., 2020).

| | wrn-40-2 wrn-16-2 | wrn-40-2 wrn-40-1 | resnet56 resnet20 | resnet110 resnet20 | resnet110 resnet32 | resnet32x4 resnet8x4 | vgg13 vgg8 |
|---|---|---|---|---|---|---|---|
| Teacher | 75.61 | 75.61 | 72.34 | 74.31 | 74.31 | 79.42 | 74.64 |
| Student | 73.26 | 71.98 | 69.06 | 69.06 | 71.14 | 72.50 | 70.36 |
| KD | 74.92 | 73.54 | 70.66 | 70.67 | 73.08 | 73.33 | 72.98 |
| FitNet | 73.58 | 72.24 | 69.21 | 68.99 | 71.06 | 73.50 | 71.02 |
| AT | 74.08 | 72.77 | 70.55 | 70.22 | 72.31 | 73.44 | 71.43 |
| SP | 73.83 | 72.43 | 69.67 | 70.04 | 72.69 | 72.94 | 72.68 |
| CC | 73.56 | 72.21 | 69.63 | 69.48 | 71.48 | 72.97 | 70.71 |
| VID | 74.11 | 73.30 | 70.38 | 70.16 | 72.61 | 73.09 | 71.23 |
| RKD | 73.35 | 72.22 | 69.61 | 69.25 | 71.82 | 71.90 | 71.48 |
| PKT | 74.54 | 73.45 | 70.34 | 70.25 | 72.61 | 73.64 | 72.88 |
| AB | 72.50 | 72.38 | 69.47 | 69.53 | 70.98 | 73.17 | 70.94 |
| FT | 73.25 | 71.59 | 69.84 | 70.22 | 72.37 | 72.86 | 70.58 |
| FSP | 72.91 | 0.00 | 69.95 | 70.11 | 71.89 | 72.62 | 70.23 |
| NST | 73.68 | 72.24 | 69.60 | 69.53 | 71.96 | 73.30 | 71.53 |
| CRD | 75.48 | 74.14 | 71.16 | 71.46 | 73.48 | 75.51 | 73.94 |
| **Ours** | **75.96** | **74.75** | **71.44** | **71.51** | **73.80** | **75.92** | **74.40** |

Table 13: **Distillation experiment with different architectures (Tian et al., 2020):** Top-1 accuracy (%) on CIFAR-100. The student models were trained with a teacher of different architecture. We report average over 3 runs as in (Tian et al., 2020).

|         | vgg13 MobileNetV2 | ResNet50 MobileNetV2 | ResNet50 vgg8 | resnet32x4 ShuffleNetV1 | resnet32x4 ShuffleNetV2 | wrn-40-2 ShuffleNetV1 |
|---------|-------------------|----------------------|---------------|-------------------------|-------------------------|-----------------------|
| Teacher | 74.64 | 79.34 | 79.34 | 79.42 | 79.42 | 75.61 |
| Student | 64.60 | 64.60 | 70.36 | 70.50 | 71.82 | 70.50 |
| KD      | 67.37 | 67.35 | 73.81 | 74.07 | 74.45 | 74.83 |
| FitNet  | 64.14 | 63.16 | 70.69 | 73.59 | 73.54 | 73.73 |
| AT      | 59.40 | 58.58 | 71.84 | 71.73 | 72.73 | 73.32 |
| SP      | 66.30 | 68.08 | 73.34 | 73.48 | 74.56 | 74.52 |
| CC      | 64.86 | 65.43 | 70.25 | 71.14 | 71.29 | 71.38 |
| VID     | 65.56 | 67.57 | 70.30 | 73.38 | 73.40 | 73.61 |
| RKD     | 64.52 | 64.43 | 71.50 | 72.28 | 73.21 | 72.21 |
| PKT     | 67.13 | 66.52 | 73.01 | 74.10 | 74.69 | 73.89 |
| AB      | 66.06 | 67.20 | 70.65 | 73.55 | 74.31 | 73.34 |
| FT      | 61.78 | 60.99 | 70.29 | 71.75 | 72.50 | 72.03 |
| NST     | 58.16 | 64.96 | 71.28 | 74.12 | 74.68 | 74.89 |
| CRD     | **69.73** | 69.11 | 74.30 | 75.11 | 75.65 | 76.05 |
| Ours    | 69.14 | **69.45** | **74.46** | **75.66** | **76.40** | **76.61** |

