# OpenReview forum: "Knowledge distillation via softmax regression representation learning"
_ICLR.cc/2021/Conference — ICLR 2021 Poster_

### Official Review · AnonReviewer2 · 2020-10-28
**Official Blind Review #2**

**Rating:** 6
**Confidence:** 4

**Review:**

#####################################################################################
Summary:

This paper proposes a new formulation of knowledge distillation (KD) for model compression. Different from the classic formulation that matches the logits between student and teacher models, this paper suggests to match the output features of the penultimate layers between student and teacher models, based on L2 distance. Two complementary variants are introduced, with one directly minimizing the distance of the original feature vectors and with the other minimizing the distance of the feature vectors projected on the teacher classifiers. The approach is evaluated in a variety of scenarios, such as different network architectures, teacher-student capacities, datasets, and domains, and compared with state-of-the-art results.

####################################################################################
Pros:

The proposed knowledge distillation formulation is simple. The paper is well written. Experimental evaluations clearly demonstrate the effect by directly matching the output features of the penultimate layers.

####################################################################################
Cons:

- While the proposed approach is interesting, its novelty seems limited, with formulations being special cases of existing methods. In particular, as the authors also mentioned, the proposed L_FM loss is a simplified FitNet loss in Romero et al., which focuses only on matching the final representation without considering the intermediate representations. The proposed L_SR loss is similar to the standard KD formulation in Hinton et al., with the only difference that the pre-trained teacher’s classifier is used for both teacher and student models.

- From the result tables, the performance improvement of the proposed approach is marginal, which is on par with existing works.

- As the authors mentioned, the L_SR loss is inspired by that only using the L_FM loss ignores the inter-channel dependencies of the feature representations h^S and h^T. So L_SR-CE is introduced. However, empirically, L_SR-CE is worse than L_SR-L2 which directly minimizes the distance between projected features. It seems that the empirical formulation is inconsistent with the motivation.

- In Eq 8, I assume that a separate classifier is also trained for the student model using the L_CE loss. How is the performance if we completely remove the student classifier? That is, the student model only consists of a feature extractor, and during inference time, we directly insert the pre-trained teacher classifier on top of the student model.

- It would be interesting to show the results when h^T and h^S have different dimensionality.

- In the ablation studies, the authors investigated “where should be losses be applied”. How is the classifier generated for the intermediate layers?

- How does the performance change with respect to different settings of the hyper-parameters alpha and beta?

- The notation is inconsistent throughout the paper. For example, h^T and h^S are used in the first half of the method section, while h_T and h_S are used later.

#################################################################################### Updated:

The authors' rebuttal addressed my concerns and I lean toward acceptance.

---

> ### Author Response · Authors · 2020-11-23
> **Response to reviewer 2 (Part1)**
>
> $\textbf{Q2.1}$: While the proposed approach is interesting, its novelty seems limited, with formulations being special cases of existing methods. In particular, as the authors also mentioned, the proposed $L_{FM}$ loss is a simplified FitNet loss in Romero et al., which focuses only on matching the final representation without considering the intermediate representations. The proposed $L_{SR}$ loss is similar to the standard KD formulation in Hinton et al., with the only difference that the pre-trained teacher’s classifier is used for both teacher and student models.
>
> $\textbf{R2.1}$: We respectfully disagree with the Reviewer on the above. Simplicity is one of the strengths of our method. Furthermore, the fact that our method is simple does not mean that it lacks novelty. Firstly, $L_{FM}$ loss has not been used anywhere else in other distillation papers including the original FitNet paper by Romero et al. Secondly, regarding the difference between $L_{SR}$ and $L_{KD}$, we emphasize that, similarly to many other papers, the devil is in the details: the difference is a fundamental one (as also confirmed by the experimental results) and by no means trivial as this simple idea has been missed previously in literature. Finally, the motivation behind our work and KD is different: our method focuses on improving the representation capacity of the penultimate layer’s output feature.
>
> $\textbf{Q2.2}$: From the result tables, the performance improvement of the proposed approach is marginal, which is on par with existing works.
>
> $\textbf{R2.2}$: We respectfully disagree on the above assessment of the reviewer. Actually, especially on ImageNet, which is the most important experiment, our method outperforms the second best (CRD) quite significantly.  Moreover, we report large improvements for real-to-binary distillation. Below we provide the data. Note that these improvements should be judged relatively to what other methods improve upon their main competitors (e.g. check OFD vs KD or CRD vs OFD on ImageNet).
>
> On CIFAR-100: Out of the 12 comparisons made on CIFAR-100 (See Tables 11 and 12), our method achieves the best performance on 12 out of 13 times:
>
> +0.38, +0.61, +0.28, +0.05, +0.42, +0.41, +0.46, -0.59, +0.34, +0.16, +0.55, +0.75, +0.56
>
> Some of these improvements are marginal (0.05, 0.16…) but most are not.
>
> On ImageNet, Top1:
>
> +0.56, +1.09 improvement over CRD.
>
> On real-to-binary distillation: +1.72% on CIFAR-100, +1.32% on ImageNet over CRD.
>
> $\textbf{Q2.3}$: As the authors mentioned, the $L_{SR}$ loss is inspired by that only using the $L_{FM}$ loss ignores the inter-channel dependencies of the feature representations $h^S$ and $h^T$. So $L_{SR-CE}$ is introduced. However, empirically, $L_{SR-CE}$ is worse than $L_{SR-L2}$ which directly minimizes the distance between projected features. It seems that the empirical formulation is inconsistent with the motivation.
>
> $\textbf{R2.3}$: There is a misunderstanding here:  $L_{SR-CE}$ and $L_{SR-L2}$ have exactly the same properties. The only difference is in the exact functional form of the loss. Other than that whatever holds for $L_{SR-CE}$ also holds for $L_{SR-L2}$.
>
> $\textbf{Q2.4}$: In Eq 8, I assume that a separate classifier is also trained for the student model using the $L_{CE}$ loss. How is the performance if we completely remove the student classifier? That is, the student model only consists of a feature extractor, and during inference time, we directly insert the pre-trained teacher classifier on top of the student model.
>
> $\textbf{R2.4}$: Thank you for the suggestion, it is an interesting experiment that we didn't consider. On ImageNet, using ResNet34 as teacher, and ResNet18 as student, our paper’s result is 71.73 whereas using your suggestion gives 71.81 which is basically the same.

---

> ### Author Response · Authors · 2020-11-23
> **Response to reviewer 2 (Part2)**
>
> $\textbf{Q2.5}$: It would be interesting to show the results when $h^T$ and $h^S$ have different dimensionality.
>
> $\textbf{R2.5}$: This is already the case for most experiments. For example, for the following pairs (T:WRN16_4 S:WRN16_2), (T:ResNet34 S:WRN16_2), (T:ResNet34 S:MobileNetV2), (T:WRN40_4 S:MobileNetV2) from CIFAR-100 and (T:ResNet50 S:MobileNet) from ImageNet, $h^T$ and $h^S$ have different dimensionality. This difference in dimensionality is adjusted with function r(.) (mentioned in the end of page 2, and beginning of page 3) which is a 1x1 convolution. Thank you for this, we will make it clearer in the paper.
>
> $\textbf{Q2.6}$: In the ablation studies, the authors investigated “where should be losses be applied”. How is the classifier generated for the intermediate layers?
>
> $\textbf{R2.6}$: There is no classifier generated for intermediate layers. For the last conv. layer of each stage (i.e. once for each resolution), we just apply Adaptive Instance Normalization to transfer the channel-wise mean of the student’s feature tensor to the teacher’s feature tensor.
>
> $\textbf{Q2.7}$: How does the performance change with respect to different settings of the hyper-parameters alpha and beta?
>
> $\textbf{R2.7}$: Thank you for pointing this out, this was an omission. We did only a basic search to find the best hyper-parameters. First, we fix $\alpha$ and search for the best $\beta$. Then, we used the best $\beta$, and search for the best $\alpha$. This is suboptimal compared to a full grid search over $\alpha$ and $\beta$. Furthermore, we searched only for 2 values: 1 and 5. Notably, we found that for all teacher-student pairs but (T:WRN40_4 S:MobileNetV2)  $\alpha=1$ is the optimal value. Furthermore, on ImageNet, the optimal values were $\alpha=1$, $\beta=1$ for all teacher-student pairs considered. We will add this information to the paper.
>
> $\textbf{Q2.8}$: The notation is inconsistent throughout the paper. For example, $h^T$ and $h^S$ are used in the first half of the method section, while $h_{T}$ and $h_{S}$ are used later.
>
> $\textbf{R2.8}$: Thank you for pointing out this! We will fix it.

---

### Official Review · AnonReviewer3 · 2020-10-29
**Review of the paper "Knowledge distillation via softmax regression representation learning"**

**Rating:** 6
**Confidence:** 4

**Review:**

**Paper summary**
This paper proposes a new knowledge distillation method by enhancing the student network's representation learning. The proposed framework is very simple: use the teacher's final fully-connected layer (or, 'projection matrix $W$' in the paper) to obtain both the teacher's and the student's logit output and optimize the student network by minimizing the gap between two logits. Experiments cover various network architectures on CIFAR and ImageNet datasets and show improvements over the other competitive knowledge distillation methods.

**Strengths**
1. The main idea to share the teacher's projection matrix $W$ for both the student and teacher's penultimate features seems new.
2. Proposed method is simple and easy to implement.
3. Experiments include a wide range of network architectures and datasets.

**Weaknesses**
1. Definition of representation learning is unclear.
The paper claims that using the pre-trained and frozen projection matrix brings decoupling of "representation learning" and "classification". Also, the paper says "Because direct feature matching might be difficult due to the lower representation capacity of the student ..." on page 2.
After reading this section, I got the following questions: 1) According to the paper's solution, the teacher's projection matrix is used to solve the lack of representation capacity of the student. Does the "representation capacity" of a network only exist in the final fully-connected layer (projection matrix)? If true, references or pieces of evidence for this should be provided. 2) Why the method of utilizing the teacher's projection matrix is called representation learning?

2. On page 4, by comparing eq (5) and eq (6), the paper claims that the KD loss $L_{KD}$ might have worse generalization ability than $L_{SR}$ since the KD loss simultaneously updates the feature extractor and classifier. The paper only shows empirical results where $L_{SR}$ works better than $L_{KD}$. This argument is not easily accepted for me, so more theoretical backup is necessary.

3. Study on the hyper-parameter $\alpha$ and $\beta$ on eq (8) is missing

4. The main idea ("softmax representation learning", or $L_{SR}$) is very simple and seems to be easily applied with other KD methods beyond FitNets method. Combining $L_{SR}$ with other KD methods (e.g, AT, OFD, etc) will verify the generalization and applicability of the proposed method.


**Post-comments to the author's response**
- Replies for Q3 and Q4 are good.
- However, the responses for Q1 and Q2 did not address my concern well.
- For Q1, the authors claim that the representation capacity is in the CNN features not in the fully-connected layer. Then what is the meaning of utilizing the teacher's projection matrix for better representation learning?
- Overall, I still this paper has more strengths than weaknesses, so I will keep my rating (6)

---

> ### Author Response · Authors · 2020-11-23
> **Response to reviewer 3**
>
> $\textbf{Q3.1}$: Definition of representation learning is unclear. The paper claims that using the pre-trained and frozen projection matrix brings decoupling of "representation learning" and "classification". Also, the paper says "Because direct feature matching might be difficult due to the lower representation capacity of the student ..." on page 2. After reading this section, I got the following questions: 1)  According to the paper's solution, the teacher's projection matrix is used to solve the lack of representation capacity of the student. Does the "representation capacity" of a network only exist in the final fully-connected layer (projection matrix)? If true, references or pieces of evidence for this should be provided. 2) Why the method of utilizing the teacher's projection matrix is called representation learning?
>
> $\textbf{R3.1}$: If we think of a CNN as a feature extraction backbone  followed by a classifier, then the "representation capacity" is related to the feature extraction part.  More specifically, representation learning, in our paper, refers to a training procedure which focuses on improving the representational capacity of the output of the penultimate layer of the student, i.e. the feature just before the student’s linear layer, denoted as $h^S$ in our paper.
>
> 1) The term representation capacity refers to that feature ($h^S$) and not the final fully-connected layer (projection matrix). This definition has been used widely in recent work on self-supervised learning [1, 2]
>
> 2) Because the  teacher's projection matrix remains frozen, applying $L_{SR}$ loss is directly focusing on optimizing $h^S$.
>
> [1] Wu, Z., Xiong, Y., Yu, S. X., and Lin, D. Unsupervised featurelearning via non-parametric instance discrimination. In CVPR. 2018
>
> [2] Chen, T., Kornblith, S., Norouzi, M., Hinton, G.: A simple framework for contrastive learning of visual representations, In ICML. 2020.
>
> $\textbf{Q3.2}$: On page 4, by comparing eq (5) and eq (6), the paper claims that the KD loss $L_{KD}$ might have worse generalization ability than $L_{SR}$ since the KD loss simultaneously updates the feature extractor and classifier. The paper only shows empirical results where $L_{SR}$ works better than $L_{KD}$. This argument is not easily accepted for me, so more theoretical backup is necessary.
>
> $\textbf{R3.2}$: Unfortunately, we do not have a theoretical justification for this claim. However, in addition to the improved accuracy results obtained by our method,  we do have an experiment which provides further empirical evidence: Section 4 -- Transferability of representations, where we compare the representational power of the student’s feature $h_s$ trained with KD and with our method. Clearly, our method can train a more powerful feature representation which generalizes better when employed on a different dataset (trained on CIFAR; tested on STL10).
>
> $\textbf{Q3.3}$: Study on the hyper-parameter α and β on eq (8) is missing
>
> $\textbf{R3.3}$: Thank you for pointing this out, this was an omission. We did only a basic search to find the best hyper-parameters. First, we fix $\alpha$ and search for the best $\beta$. Then, we used the best $\beta$, and search for the best $\alpha$. This is suboptimal compared to a full grid search over $\alpha$ and $\beta$. Furthermore, we searched only for 2 values: 1 and 5. Notably, we found that for all teacher-student pairs but (T:WRN40_4 S:MobileNetV2) $\alpha=1$ is the optimal value. Furthermore, on ImageNet, the optimal values were $\alpha=1$, $\beta=1$ for all teacher-student pairs considered. We will add this information to the paper.
>
> $\textbf{Q3.4}$: The main idea ("softmax representation learning", or $L_{SR}$) is very simple and seems to be easily applied with other KD methods beyond FitNets method. Combining $L_{SR}$ with other KD methods (e.g, AT, OFD, etc) will verify the generalization and applicability of the proposed method.
>
> $\textbf{R3.4}$: In Table 10, appendix A.2, we already provide some results by combining our method with KD and AT. As it can be seen adding KD slightly hurts performance as expected. From the same table, we observe that AT provides very low improvements compared to our method and hence combining the two methods works worse than our method alone. As also suggested by you, during rebuttal time, we ran the experiment OFD+ours where we get: 75.90,  79.17, $\textbf{79.39}$,  69,90,  72.50,  77.79,  $\textbf{75.65}$, 71.56, $\textbf{71.96}$ where bold indicates improvement over our method alone. This shows some promising direction, however a more comprehensive investigation is needed to fully realise the potential of combining the two methods together.

---

### Official Review · AnonReviewer1 · 2020-10-29
**A simple yet effective form of loss for general knowledge distillation**

**Rating:** 7
**Confidence:** 5

**Review:**

This paper proposed a novel method for knowledge distillation. The idea is to utilize the teacher’s pre-trained classifier to train the student’s penultimate layer feature by adopting two losses: (a) the Feature Matching loss LFM and (b) the Softmax Regression loss LSR. The latter is designed to take into account the classification task at hand, which imposes that for the same input image, the teacher’s and student’s feature produce the same output when passed through the teacher’s pre-trained and frozen classifier.

The whole idea is easy and clear. The authors also provided necessary experiments to rationalize the approach, for example, why the penultimate layer? why student classifier not included? The final model is thoroughly compared with state-of-the-arts knowledge distillation methods. Despite in several tasks, the proposed model fails to succeed the CRD method, given the simplicity of the proposed method and its superiority over most of the tasks, I think the results are satisfactory.

Still, I would like to give a suggestion:
1.	After CRD, proposed in 2019, there appears many new methods with refreshing performance. For example, [1] Xu G , Liu Z , Li X , et al. Knowledge Distillation Meets Self-Supervision[C]. in ECCV 2020. It would be more persuasive if comparisons with more recent methods could be provided.

---

> ### Author Response · Authors · 2020-11-23
> **Response to reviewer 1**
>
> Thank you for your positive feedback.
>
> $\textbf{Q1.1}$: After CRD, proposed in 2019, there appears many new methods with refreshing performance. For example, [1] Xu G , Liu Z , Li X , et al. Knowledge Distillation Meets Self-Supervision[C]. in ECCV 2020. It would be more persuasive if comparisons with more recent methods could be provided.
>
> $\textbf{R1.1}$: Thank you for providing this very interesting paper which we will cite and discuss in our paper. The paper follows a different path to KD by proposing a self-supervised task whereas our method uses the more traditional supervised learning approach. Hence, we believe that the improvements reported in [1] could be complementary to ours. In fact, one of the deployed losses in [1] is standard KD which can be replaced by our method. So during rebuttal time we used the authors' provided code [2], and replaced $L_{KD}$ in Eq. (8) of [1]  with our proposed losses $L_{FM}$ and $L_{SR}$, and trained the following Teacher-Student pair (T:WRN40_2 S:WRN16_2) on CIFAR-100. We were able to obtain +0.43% improvement. This result, which we are confident that can be further improved, is indicative of the effectiveness of the potential synergy between the two approaches.
>
> Finally, although we will of course discuss and cite the aforementioned paper, we would also like to bring up ICLR guidelines: "if a paper was published on or after Aug 2, 2020, authors are not required to compare their own work to that paper."
>
> [1] Xu G , Liu Z , Li X , et al. Knowledge Distillation Meets Self-Supervision[C]. In ECCV 2020
>
> [2] https://github.com/xuguodong03/SSKD

---

### Official Review · AnonReviewer4 · 2020-11-06
**Simple and effective methods**

**Rating:** 7
**Confidence:** 4

**Review:**

This paper propose to leverage the frozen classifier from the teacher model for training better representations for the student model. By further combining with another feature matching loss, the proposed methods outperforms previous methods (such as KD, AT, CRD) on many benchmarks.

Pros:
- The proposed method is very simple, and effective.
- Authors demonstrate the effectiveness of this method, on CIFAR-10, CIFAR-100, ImageNet, as well as transferring to STL-10. They also apply real-to-binary distillation, which is interesting.
- The paper is well-written and very easy to follow.

Overall, I did not see much disadvantages of this proposed method. I am a bit surprised that such simple method has been ignored before. But some parts that can be further enhanced is:
- Limitation: despite its simplicity, this method, in its current format, assumes that the teacher classifier is available, which is not always true in may applications. Besides, it's not clear that whether this method is still effective, if the teacher model is not a standard classification model, e.g., if you want to compress a regression model, whether you can still use teacher's frozen regressor to guide student representation is still unknown. So I encourage the authors to add a section to discuss it.

Overall, I like this simple and surprisingly effective method.

=== update ===

The provided response is reasonable and helps address some of my concerns. I would keep my rating as acceptance.

---

> ### Author Response · Authors · 2020-11-23
> **Response to reviewer 4**
>
> Thank you for the positive feedback regarding our work.
>
> $\textbf{Q4.1}$: this method, in its current format, assumes that the teacher classifier is available, which is not always true in may applications. Besides, it's not clear that whether this method is still effective, if the teacher model is not a standard classification model, e.g., if you want to compress a regression model, whether you can still use teacher's frozen regressor to guide student representation is still unknown. So I encourage the authors to add a section to discuss it.
>
> $\textbf{R4.1}$: Most papers that we’re comparing with, including classic papers like KD [2], AT[3] but also state-of-the-art methods like RKD [4], PKT [5], a Teacher is always available. Furthermore, we primarily focused on classification and all the papers we are comparing with are also focusing on classification problems. Furthermore, it is not clear how KD can be applied for regression tasks. This is because one of the key features of knowledge distillation is using the soft outputs of the teacher as supervisory signals for supervised learning which capture inter-class dependencies (as opposed to using one-hot labels). For regression, there are no such signals as the network outputs the regressed values only. On the contrary, we can argue that since our method focuses on improving the penultimate layer's output feature, it can be effective for regression tasks too. So, we did follow your advice and applied our method to a regression problem: facial landmark localisation. Given a facial image the task is to localise a set of facial landmarks in terms of (x,y) coordinates which can be solved by using a conv. net to directly regress the (x,y) coordinates of the facial landmarks.  We used the WFLW [1] dataset, which is one of the hardest benchmarks for this task. Performance is measured in terms of Normalised Mean Error (lower is better), the standard metric for the problem. In our experiment, we used a ResNet50 as the teacher and a ResNet8 as the student. The results in the following table confirm our hypothesis.
>
> | Teacher|Student|KD(with L2 loss)|AT| RKD|PKT|$L_{FM}$|Ours|
> |:----------:|:----------:|:----------:|:----------:|:----------:|:----------:|:----------:|:----------:|
> |6.38|7.43|7.32|6.96|6.94|7.09|7.14|6.81|
>
> [1] Wu, Wayne, et al. "Look at boundary: A boundary-aware face alignment algorithm." In  CVPR. 2018.
>
> [2] Geoffrey Hinton, Oriol Vinyals, and Jeff Dean. Distilling the knowledge in a neural network. In  arXiv. 2015.
>
> [3] Sergey Zagoruyko and Nikos Komodakis. Paying more attention to attention: Improving the performance of convolutional neural networks via attention transfer. In ICLR. 2017.
>
> [4] Wonpyo Park, Dongju Kim, Yan Lu, and Minsu Cho. Relational knowledge distillation. In CVPR. 2019.
>
> [5] Passalis, Nikolaos, and Anastasios Tefas. "Learning deep representations with probabilistic knowledge transfer." In ECCV. 2018.

---

### Decision · Program_Chairs · 2021-01-07
**Final Decision**

**Decision:**

Accept (Poster)

**Comment:**

This paper proposes a new idea for performing knowledge distillation by leveraging teacher’s classifier to train student’s penultimate layer feature via proposing suitable loss functions. Reviewers appreciate the simultaneous simplicity and effectiveness of the method. A comprehensive set of studies are performed to empirically show the effectiveness of the method. Specifically, the proposed distillation method is shown to outperform state-of-the-art across various network architectures, teacher-student capacities, datasets, and domains. The paper is well-written and is easy to follow. All reviewers rate the paper on the accept side (after the rebuttal) and believe the new perspective this work provides on distillation and its simplicity to implement can lead it to gain high impact. I concur with the reviewers and find this submission a convincing empirical work, and thus recommend for accept.